# LncRNA BCYRN1 as a Potential Therapeutic Target and Diagnostic Marker in Serum Exosomes in Bladder Cancer

**DOI:** 10.3390/ijms25115955

**Published:** 2024-05-29

**Authors:** Junya Arima, Hirofumi Yoshino, Wataru Fukumoto, Ichiro Kawahara, Saeki Saito, Gang Li, Ikumi Fukuda, Sayaka Iizasa, Akihiko Mitsuke, Takashi Sakaguchi, Satoru Inoguchi, Ryosuke Matsushita, Masayuki Nakagawa, Shuichi Tatarano, Yasutoshi Yamada, Hideki Enokida

**Affiliations:** Department of Urology, Graduate School of Medical and Dental Sciences, Kagoshima University, Kagoshima 890-8544, Japan

**Keywords:** bladder cancer, biomarker, exosome, lncRNA, *BCYRN1*

## Abstract

Bladder cancer (BC) is a common genitourinary malignancy that exhibits silent morbidity and high mortality rates because of a lack of diagnostic markers and limited effective treatments. Here, we evaluated the role of the lncRNA brain cytoplasmic RNA 1 (*BCYRN1*) in BC. We performed loss-of-function assays to examine the effects of *BCYRN1* downregulation in T24 and BOY BC cells. We found that *BCYRN1* downregulation significantly inhibited the proliferation, migration, invasion, and three-dimensional spheroid formation ability and induced apoptosis in BC cells. Additionally, gene set enrichment analysis (GSEA) using RNA sequences from tumor fractions showed that *BCYRN1* downregulation decreased the expression of mRNAs associated with the cell cycle. These findings were supported by observations of G_2_/M arrest in flow cytometry assays. Finally, we examined the expression of serum exosomal *BCYRN1* as a biomarker. Clinically, *BCYRN1* expression in serum exosomes from patients with BC (*n* = 31) was significantly higher than that in healthy donors (*n* = 19; mean difference: 4.1-fold higher, *p* < 0.01). Moreover, in patients who had undergone complete resection of BC, serum exosomal *BCYRN1* levels were significantly decreased (*n* = 8). Thus, serum exosomal *BCYRN1* may be a promising diagnostic marker and therapeutic target in patients with BC.

## 1. Introduction

Bladder cancer (BC) is the second most common urogenital neoplasm worldwide, with approximately 524,000 new cases diagnosed and 229,000 BC-related deaths in 2019 [1]. BC was also the 12th most common newly diagnosed cancer type in 2020 and the sixth most common cancer in men [2]. Approximately 70–80% of patients with BC are diagnosed with non-muscle-invasive bladder cancer (NMIBC) with a “superficial” T stage (Tis, Ta, or T1). The remaining cases are diagnosed with muscle-invasive bladder cancer (MIBC). Notably, screening for BC through the detection of microscopic hematuria has demonstrated a 20% reduction in the rate of MIBC and a decline in disease-specific mortality compared with standard care practices [3]. However, the prognosis for NMIBC is poor, and 21% of cases progress to MIBC [4]. Moreover, advanced BC, including MIBC and BC with metastasis, has a high mortality rate. The management of advanced BC has improved with the introduction of novel treatment strategies, including the development of immune checkpoint inhibitors [5,6,7], antibody-drug conjugates [8], and fibroblast growth factor receptor inhibitors [9,10]. However, the effectiveness of these treatments is limited because of the acquisition of chemotherapy resistance. In patients with NMIBC, transurethral resection of bladder tumor (TURBT) is the first curative or diagnostic intervention. After TURBT, patients with pathological high-grade tumors or cT1 NMIBC typically undergo a second transurethral resection (TUR) to remove the residual tumor, with the goal of prolonging overall survival [11,12]. Surveillance or active treatments, such as Bacille Calmette-Gueri treatment, intravesical chemotherapy, or total cystectomy, are selected for other patients [13]. Patients with BC who do not require total cystectomy are followed up using both urinary cytology and transurethral cystoscopy. The sensitivity of urinary cytology for localized BC remains suboptimal, although the specificity of urinary cytology is high. Despite choosing the most effective approach according to risk factors, Sven van den Bosch et al. reported that 21% of patients with NMIBC progress to MIBC, and 14% of patients die of BC [4]. Thus, there is an urgent need for a highly sensitive and inexpensive diagnostic approach to enhance the detection of early-stage or locally recurrent BC.

Noncoding RNAs (ncRNAs) account for 90% of RNAs transcribed from the human genome [14]. Although they do not encode proteins, ncRNAs have important functional roles in cells, including cancer cells. ncRNAs are classified according to length [14]; ncRNAs consisting of 20–25 nucleotides are called microRNAs, whereas those consisting of more than 100–200 nucleotides are called long noncoding RNAs (lncRNAs). We have reported the potential of microRNAs as prognostic markers and in therapeutic applications [15,16,17]. Several tumor-suppressive microRNAs, including the *miR-1*/*TAGLN2* axis [18], *miR-34a*/*HOTAIR* axis [19], *miR-195*/*-497* cluster [20], and *miR-199* family/*ITGA3* axis [21], may have therapeutic applications. Additionally, urinary *miR-96*/*-183* may be a diagnostic marker [22]. However, the methods used to detect urinary microRNAs and apply them as diagnostic markers are limited by the potential for urinary tract infection and blood cell contamination. Therefore, it is necessary to develop novel methods for prognostic marker analysis and therapeutic applications.

Extracellular vesicles (EVs) have attracted attention as a means of communication with surrounding cells or with distant cells using the bloodstream [23]. EVs have recently been used in the search for biomarkers of cancer detection [24,25,26] and as a drug delivery system [27,28,29]. EVs secreted from the cell membrane contain proteins, mRNAs, microRNAs, lncRNAs, circular RNAs, metabolites, lipids, and other small substances that can be detected in the cytoplasm of the original cell. EVs in the 50–150 nm range are called exosomes, whereas those in the 100–1000 nm range are called microvesicles. Cancer cells secrete a greater number of exosomes than normal cells, and cancer-derived exosomes have a strong potential to change both regional and distant microenvironments [30,31,32]. In a previous report, we showed that exosomal *miR-1* reduced the viability of cells treated with the exosomes. Moreover, we also reported that exosomes from the sera of patients with renal cell cancer exhibited significant upregulation of *MYO15A* compared with that of healthy controls [33]. Thus, exosomes collected by liquid biopsy (blood, ascites, urine, etc.) have the potential to act as therapeutic and diagnostic tools.

Recently, several lncRNAs, including brain Cytoplasmic RNA 1 (*BCYRN1*), have been reported to be associated with tumor progression, metastasis, and acquired anticancer drug resistance [34,35,36,37]. Although *BC200* is known to be highly expressed in dendrites and is thought to be involved in local translational regulation [38,39] and *BCYRN1* has been shown to be involved in epilepsy, Alzheimer’s disease, and cognitive function of multiple sclerosis [40,41,42], *BCYRN1* is known to exhibit high expression in several cancer types, such as non-small-cell lung cancer [43,44,45], gastric cancer [46,47,48], colorectal carcinoma [49,50,51], NK/T-cell lymphoma [52], and glioblastoma [53]. Additionally, abundant *BCYRN1* promotes cancer metastasis and progression by adsorbing microRNAs “like a sponge” [54]. Knockdown of the lncRNA *BCYRN1* using small interfering RNA (siRNA) reduces cell viability in several types of cancer [55]. However, few reports have described the effects of *BCYRN1* on BC cells or its potential application as a serum-based diagnostic tool for BC. Accordingly, in this study, we investigated the role of *BCYRN1* in BC cells and explored the diagnostic and therapeutic potential of serum-derived exosomal *BCYRN1* in patients with BC.

## 2. Results

### 2.1. Knockdown of BCYRN1 Decreased Cell Viability in BC Cells

First, to investigate the effects of *BCYRN1* on BC cells, the BC cells were transfected with siRNAs, and loss-of-function assays were performed. Transfection with si*BCYRN1*-1/-2 significantly reduced *BCYRN1* mRNA expression (Appendix A). XTT assays revealed significant inhibition of BC cell proliferation ability upon *BCYRN1* knockdown (Figure 1A). Spheroid formation assays also demonstrated the inhibition of spheroid formation in the BC cells transfected with si*BCYRN1*-1/-2 (Figure 1B). Cell migration ability, as demonstrated using WH assays, was markedly suppressed in cells transfected with si*BCYRN1*-1/-2 compared with that in the control cells (Figure 1C). Furthermore, cell invasion ability, as demonstrated using Matrigel invasion assays, was markedly suppressed in cells transfected with si*BCYRN1*-1/-2 compared with that in control cells (Figure 1D). These results suggested that the knockdown of *BCYRN1* decreased the viability of BC cells.

### 2.2. Cell Viability Was Decreased by BCYRN1 Knockdown in a Xenograft Mouse Model

The knockdown of *BCYRN1* suppressed cell proliferation, invasion, and migration. Given that si-*BCYRN1*-1 demonstrated a more pronounced impact on knockdown efficacy, we used only si-*BCYRN1*-1 in our mouse experiments, thereby minimizing the number of casualties. Therefore, we next used a xenograft mouse model with T24 cells transfected with si-*BCYRN1*-1. The si-*BCYRN1*-1-transfected cells were subcutaneously injected into the right sides of mice (*n* = 5), whereas the siNC-transfected cells were injected into the left sides of mice (*n* = 5). The downregulation of *BCYRN1* significantly decreased tumor volume, with significant differences in tumor volume and tumor weight observed over time (Figure 2A,B). No serious adverse events were observed in this xenograft assay.

Next, Ki-67 immunostaining was performed to confirm the proliferation ability of the resected tumor cells. The results showed that the number of Ki-67-positive cells was significantly reduced in the *BCYRN1*-knockdown cells (Figure 2C).

### 2.3. BCYRN1 Downregulation Affected the Cell Cycle and Cell Apoptosis

To further investigate the role of *BCYRN1*, we performed mRNA sequencing of total RNAs extracted from T24 and BOY cells transfected with si*BCYRN1* or siNC. GSEA showed significant changes in the gene sets associated with the cell cycle (Figure 3A). The gene sets related to the G_2_/M checkpoint, E2F targets, and the mitotic spindle were significantly downregulated in the *BCYRN1*-knockdown BC cells. Western blotting assays demonstrated that there was a significant decrease in CDK2 in the si*BCYRN1*-transfected BC cells (Figure 3B), suggesting that there were abnormalities in the cell cycle. Considering the possibility that apoptosis may be related to decreased proliferative capacity, we checked the expression of cleaved PARP and Bcl-xL in the BC cells transfected with si*BCYRN1*. We confirmed an increase in cleaved PARP expression and a decrease in Bcl-xL expression in the *BCYRN1*-knockdown BC cells (Figure 3C).

### 2.4. Knockdown of BCYRN1 Induced G_2_/M Arrest and Apoptosis in BC Cells

Next, we performed cell cycle analysis using T24 and BOY cells transfected with si*BCYRN1* or siNC, since RNA-seq and the GSEA analysis showed a significant downregulation (lower *p*-value, higher gene ratio, or both) of several gene sets related to the cell cycle than other gene sets. Notably, si*BCYRN1*-transfected cells showed significant increases in the G_2_/M ratio, suggesting G_2_/M arrest, when compared with mock- and siNC-transfected cells. These data indicate that the downregulation of *BCYRN1* caused G_2_/M arrest in BC cells (Figure 4A). We also performed apoptosis assays using T24 and BOY cells transfected with si*BCYRN1*-1/-2 or siNC to confirm changes in cell behaviors related to apoptosis. The percentage of apoptotic cells significantly increased compared with that in mock- and siNC-transfected cells (Figure 4B).

### 2.5. High Serum Exosomal BCYRN1 Expression Was Observed in Patients with BC Tumors

Given that knocking down *BCYRN1* reduces the activity of bladder cancer cells, we investigated the possibility of using serum *BCYRN1* as a biomarker. We extracted exosomes from serum specimens from patients with BC (*n* = 31) and healthy donors (*n* = 19), as shown in Figure 5A. RT-PCR was then performed. The expression levels of serum exosomal *BCYRN1* in patients with untreated BC were higher than those in healthy donors (mean difference: 4.1-fold higher, *p* = 0.0085; Figure 5B). *GAPDH* (Glyceraldehyde-3-phosphate dehydrogenase) used for intrinsic control to compare the amount of serum exosomes, as we already reported [33]. However, there were no significant changes in T stage or progression-free survival between the higher *BCYRN1* expression group (*n* = 15) and the lower *BCYRN1* expression group (*n* = 16; Appendix A).

To further explore the utility of serum exosomal *BCYRN1* as a biomarker, we collected serum exosomes from patients who underwent the first and second TUR. All second TURs were performed 4–6 weeks after the first TUR according to the usual treatment sequence. Exosomal RNAs were extracted from sera to compare the expression levels of exosomal *BCYRN1* between specimens collected at the first TUR and the second TUR (*n* = 7). Eight patients showed reductions in serum exosomal *BCYRN1* at the second TUR and had no residual tumor formation (*n* = 8, *p* = 0.0082, Figure 5C,D). Conversely, one patient’s serum exosomal *BCYRN1* at the second TUR was higher than that at the first TUR, and a residual tumor was observed at the second TUR (Figure 5C,D).

## 3. Discussion

*BCYRN1*, also known as *BC200*, is a lncRNA normally expressed in neural cells. Zheng et al. reported exosomal *BCYRN1* drives lymphatic metastasis of BC [56]. On the other hand, decreased *BCYRN1* expression reduces cell activity in hepatocellular carcinoma, gastric cancer, and non-small cell lung cancer [43,46,57]. Our study showed that *BCYRN1* knockdown blocked cell proliferation, migration, and invasion ability in vitro and in vivo. In the xenograft assay, we observed a decrease in ki-67 expression in the tumor samples treated with si-*BCYRN1*, but other markers of proliferation or apoptosis are not examined in this experiment. However, the downregulation of *BCYRN1* induced apoptosis while increasing cleaved PARP expression and reducing Bcl-xL expression in vitro. Bcl-xL is a member of the Bcl-2 family, which consists of apoptosis inducers and regulators [58]. The flow cytometry assays indicated that *BCYRN1* knockdown also induced G_2_/M arrest. Together, these findings indicate that targeting *BCYRN1* may have potential therapeutic applications in patients with BC. By contrast, we observed no significant changes in cell proliferation in BC cells in which *BCYRN1* overexpression was induced using lentivirus. Additionally, there were no significant changes in overall survival between groups with and without *BCYRN1* amplification, as determined by analysis of the cBioPortal for Cancer Genomics (http://www.cbioportal.org/) using The Cancer Genome Atlas database, in accordance with the standards of the publication guidelines provided by TCGA (Appendix A). It is possible that *BCYRN1* shows naturally high expression in BC cells, explaining the lack of changes observed with overexpression. Although previous reports have shown that *BCYRN1* binds to and acts on microRNAs and proteins, we did not investigate the intracellular molecules that *BCYRN1* interacts with to produce these results. Further studies using RNA immunoprecipitation assays and other approaches are needed to elucidate the molecules to which *BCYRN1* could bind. In addition, we used siRNA, a transient suppressor, in the xenograft assay and did not compare the expression of *BCYRN1* in the tumor fragments between the si-control and si-*BCYRN1* because 31 days had passed since the assay started. Therefore, we need to pay careful attention to the results, and future work using permanent downregulation of gene expression, such as shRNA, is necessary.

Liquid biopsies, which involve the collection of internal fluids, such as blood, urine, ascites, saliva, and cervicovaginal lavage fluid, may be used for molecular profiling of cancers, enabling cancer screening and precision medicine. In recent years, liquid biopsies have been actively studied and clinically applied because of improvements in the technologies for analyzing trace specimen samples. After collecting specimens, cell-free DNA (cfDNA), circulating tumor DNA (ctDNA), circulating tumor cells, proteins, and exosomes, which are all contained in liquid samples, are analyzed to determine the conditions of cells at primary and metastatic sites [59,60,61,62,63]. Exosomes are secreted from living cells, whereas ctDNAs/cfDNAs are derived from necrotic or apoptotic cells [64]. Thus, liquid biopsy using exosomes may be more advantageous than liquid biopsy using ctDNA/cfDNA. Because tumor cell-derived exosomes contain the contents of parental cells, many studies have evaluated exosomes in serum to improve our understanding of the state of primary and metastatic tumor tissue [32,33,65,66,67]. In our study, we showed that serum exosomal *BCYRN1* expression was significantly higher in patients with BC than in healthy donors. We also observed that serum exosomal *BCYRN1* expression was markedly reduced after the complete dissection of bladder tumors, and patients who exhibited reduced *BCYRN1* expression showed no recurrence after surgery for more than 2 years. In contrast, one patient whose serum exosomal BCYRN1 was not reduced after the first TUR showed the presence of a bladder tumor on the second TUR surgery and local recurrence within one year. These findings suggested that the level of circulating exosomal *BCYRN1* may indicate the presence of a tumor in the bladder. Changes in serum exosomal *BCYRN1* with and without metastases were not considered in this study. Additionally, our samples did not show significant differences in serum exosomal *BCYRN1* expression between primary T stages (Tis, Ta, and T1 versus T2 and higher) (Appendix A). Furthermore, we did not observe any notable differences in progression-free survival and overall survival between the high/low serum exosomal *BCYRN1* expression groups (Appendix A). Given the small sample size and the fact that all patients were from a single institution, further studies using a larger patient cohort and more precise analyses are needed to determine whether *BCYRN1* expression levels affect local recurrence and distant metastasis and to confirm the results of our study. In addition, due to the lack of evidence for control genes, this study did not include specific positive and negative controls of exosome. We selected *GAPDH* as an intrinsic control to enable comparison of the expression level of serum exosomal *BCYRN1*, thereby facilitating standardization of the quantity of isolated exosomes. It should be noted that although several comparative methods using *GAPDH* have been reported, it is not yet the standard method for quantifying RNA in serum exosomes. Exosomes have also attracted attention as drug-transport mediators because of their affinity to cell membranes, owing to their dual lipid membrane structure [65,68,69]. For example, exosomes derived from BC cells have been shown to exhaust CD8+ cells and M2 macrophages in the tumor microenvironment to affect the antitumor immunity of the host [70]. Some exosomal lncRNAs have also been reported to act as cancer promotors. Chen et al. reported the role of the exosomal lncRNA *LNMAT2* in the enhancement of lymphatic metastasis in BC [71]. Additionally, Zheng et al. demonstrated the tumor-suppressive effect of the exosomal lncRNA *PTENP1* in BC [72]. To ensure that *BCYRN1* expression levels were comparable regardless of the quantity of exosomes collected, we used *GAPDH* within purified exosomes as an intrinsic control. Our study lacks consideration of those known BC aggravating factors, the RNAs included in exosomes. A more accurate diagnosis of UC may be achieved by focusing on multiple RNAs and comparing their expression levels. This is a subject for future research. Several other studies have shown that exosomes have potential as therapeutic tools for carrying anticancer drugs, RNAs, proteins, and immune regulators [73,74]. Zheng et al. previously reported that urine exosomal *BCYRN1* drives lymphatic metastasis of BC [56]. Serum exosomes isolated from patients with high *BCYRN1* expression have a potential revitalizing effect for BC. However, we could isolate only 100–200 μg of exosomes from 10 mL of the patient’s blood sample in our study. The quantity of isolated exosomes was insufficient for the experiment of co-culture assay, adding exosomes isolated from the patient’s sera. However, each urinary/serum exosomal RNA is considered to represent a different pathology of the cancer. While urine exosomes reflect the status of neoplastic lesions in the urinary tract, urinal contamination with hematuria, pyuria, and bacteriuria should be ruled out, serum exosomal samples reflect the whole-body distribution of bladder cancer and do not need to consider urinal contamination. Thus, the sample collection method may need to be modified according to the purpose when used for diagnosis of local tumors, systemic metastasis, or changes of tumoral condition after treatment; this is a subject for future research. Due to limitations of the ethical review, we were only able to collect serum from patients with bladder tumors without metastasis in this study. However, the evaluation of *BCYRN1* expression over time in the same patient has never before been reported. The limitation of our study is the small sample size. We collected only eight patients in the complete resection group, and only one patient in the residual tumor group. Prospects for this study include the collection of additional expression data from BC patients to enhance the reliability of the study’s findings. Moreover, there are also some limitations in the therapeutic use of exosomes, such as how to insert the drugs into the exosomes, and how to deliver the exosomes to the target cells. It is interesting and worth testing whether a co-culture assay with *BCYRN1*-expressing exosomes from patients can regulate non-transformed cells into cancerous types and, as a result, evaluate their effects on cell cycle analysis and cell apoptosis in future experiments.

In conclusion, the knockdown of *BCYRN1* led to decreased viability in BC cells both in vitro and in vivo. Furthermore, serum exosomal *BCYRN1* levels were higher in patients with BC than in healthy controls. Serum exosomal *BCYRN1* levels significantly decreased with complete resection of BC but did not decrease in a patient with a residual bladder tumor after resection. These findings suggest that serum exosomal *BCYRN1* may be a potential therapeutic and diagnostic target for BC.

## 4. Materials and Methods

### 4.1. Cell Culture

T24 and BOY human BC cell lines were used for this study. The BOY cells were established in our laboratory in 1986 using a specimen from a 66-year-old Asian man who underwent total cystectomy for invasive BC [75]. After the cystectomy, numerous lung metastases appeared, and despite undergoing various treatments, the patient succumbed to the disease. The creation of this cell line was carried out with the patient’s understanding and documented consent and was approved by the ethics committee of the appropriate institution. The T24 cells were purchased from the American Type Culture Collection (Manassas, VA, USA). The T24 and BOY cells were cultured in MEM medium containing 10% fetal bovine serum, 50 U/mL of penicillin, and 50 µg/mL of streptomycin in a humidified environment consisting of 95% air and 5% CO_2_ at 37 °C.

### 4.2. Transfection with siRNA

For the loss-of-function assays, the cells were transfected with the following previously reported siRNAs: *BCYRN1* siRNA-1, sense 5′-GUAACUUCCCUCAAAGCAAdTdT-3′ and antisense 5′-UUGCUUUGAGGGAAGUUACdTdT-3′ (Japan Bio Services Co., Osaka, Japan); *BCYRN1* siRNA-2, sense 5′-CGCCUGUAAUCCCAGCUCUCAdTdT-3′ and antisense 5′-UGAGAGCUGGGAUUACAGGCGdTdT-3′ (Japan Bio Services Co.); and negative control siRNA (si-NC; D-001810-10; Dharmacon; Horizon Discovery Group plc, Cambridge, UK). For silencing *BCYRN1*, cells (1 × 10^5^ mL^−1^) were transfected with 10 nM of siRNA using Lipofectamine RNAiMAX transfection reagent (Thermo Fisher Scientific, Inc., Waltham, MA, USA) and Opti-MEM (Thermo Fisher Scientific, Inc.), as previously reported [76]. After 48 h, the cells were harvested and used in the subsequent experiments.

### 4.3. RNA Extraction and Reverse Transcription Polymerase Chain Reaction (RT-PCR)

For total RNA extraction from the cells, ISOGEN (Nippon Gene, Tokyo, Japan) was used to prepare lysates of cultured cells according to the manufacturer’s instructions. From serum-derived exosomes, a miRNeasy mini kit (Qiagen, Redwood City, CA, USA) was used to isolate total RNA. The RNA concentration was measured spectrophotometrically, and its quality was assessed using an Agilent 2100 Bioanalyzer (Agilent Technologies, Santa Clara, CA, USA). The cDNA was synthesized from 250 or 110 ng of RNA (for RNA from cells or exosomes, respectively) using a High Capacity cDNA Reverse Transcription Kit (Applied Biosystems, Foster City, CA, USA). A SYBER green quantitative PCR-based array approach was applied using the primer sets mentioned above. The specificity of the amplification was monitored using the dissociation curve of the amplified product. Gene expression levels relative to *GUSB* or *GAPDH* (for RNA from cells or exosomes, respectively) were calculated using the 2^−ΔΔCT^ method. The following primer sets were used to measure the mRNA expression levels in RT-PCR: *BCYRN1*, forward primer 5′-GCCTGTAATCCCAGCTCTCA-3′ and reverse primer 5′-GGTTGTTGCTTTGAGGGAAG-3′; *GAPDH*, forward primer 5′-GGAGCGAGAATCCCTCCAAAA-3′ and reverse primer 5′-GGCTGTTGTCATACTTCTCA-3′; and *GUS* Beta, forward primer 5′-TTGCTCACAAAGGTCACAGG-3′ and reverse primer 5′-CGTCCCACCTAGAATCTGCT-3′.

### 4.4. RNA Sequencing and Gene Set Enrichment Analysis (GSEA)

Total RNA extracted as described above was subjected to mRNA sequencing performed by Riken Genesis Corp. (Tokyo, Japan). The library was prepared by adding adapters to the fragmented RNA samples. The length of the library was 303–314 bp. Sequencing of the formed clusters in the S4 flow cell was performed using NovaSeq 6000 (Illumina, Inc., San Diego, CA, USA), a next-generation sequencer. The effective read length was 100 bp, and the analysis was performed using the paired or multiplex method. Gene expression data analysis was carried out using GSEA v4.3.2 for Windows with the Individual Human Gene Set GMTs from the Molecular Signatures Database, using gene set collections categorized under Hallmark genes v2023.2. The gene expression dataset was derived from RNA sequencing. GSEA was executed in the collapse mode, and enrichment scores were computed, considering the weight associated with each gene.

### 4.5. Cell Proliferation, Three-Dimensional (3D) Spheroid Formation, Migration, and Invasion Assays

XTT assays were performed to measure the cell proliferation ability using an XTT kit (Roche Diagnostics, Basel, Switzerland). BC cells (1.0 × 10^3^ cells/well) were cultured in 96-well plates. After 72 h, the cells were treated with 20 µL of XTT reagent and incubated in a 5% CO_2_ incubator at 37 °C. The plate was read 2 h later at 450 nm using a microplate reader.

Spheroid assays were performed to measure cell proliferation ability in spheroids. BC cells (3.0 × 10^3^ cells/well) were cultured in a Cell-able 96-well plate (TOYO GOSEI, Chiba, Japan). After 72 h, random-site micrographs of the spheroids were obtained. To evaluate proliferation ability, the cells were fluorescently stained with Hoechst 33342 (Dojindo, Kumamoto, Japan; 24 µL/well), and fluorescence was evaluated (excitation wavelength: 350 nm; emission wavelength: 461 nm) using an Infinite 200 Pro microplate reader (TECAN Trading AG, Männedorf, Switzerland).

Wound healing (WH) assays were performed to measure cell migration activity. BC cells (1.0 × 10^5^ cells/well) were cultured in 6-well plates. After 48 h, 3 wounds were created in the cell monolayer in each well using sterilized P1000 micropipette tips. The initial gap length and the remaining gap length after 12 h were calculated from micrographs. Four micrographs of random fields of the wounds were used for quantification.

Transwell invasion assays were performed to measure cell invasion activity. Matrigel invasion chambers (Corning Biocort, Bedford, MA, USA) with bottoms made from PET membranes (8.0 µm pores) and coated with a thin layer of Matrigel basement membrane matrix were used as cell culture inserts in 24-well tissue culture companion plates. BC cells (1.0 × 10^5^ cells/well) were cultured on the cell culture inserts. After 24 h, cells that had passed through the micropores and adhered to the surface of the chamber were imaged at six random sites and counted.

### 4.6. Apoptosis and Cell Cycle Analysis

For fluorescent-activated cell sorting, each of 1.0 × 10^5^/well cells was transfected with 10 nM of siRNA for 48 h using the methodology described in Section 4.2 and collected for subsequent processing. Apoptosis assays were performed by double staining with fluorescein isothiocyanate (FITC)-Annexin V and propidium iodide using a FITC-Annexin V Apoptosis Detection Kit (BD Biosciences, Franklin Lakes, NJ, USA) and flow cytometry (CytoFLEX Analyzer; Beckman Coulter, Brea, CA, USA). CytExpert 2.3.0.84 software (Beckman Coulter) was used to classify the cells into four categories: viable cells, dead cells, early apoptotic cells, and apoptotic cells. Cell cycle analysis was performed by single staining with a Cycletest Plus DNA Reagent Kit (BD Biosciences, San Diego, CA, USA) after the cells were fixed for 96 h in 70% ethanol and washed twice with phosphate-buffered saline. DNA content was determined by taking the integrated intensity of each cell’s fluorescent signal using a CytoFLEX analyzer. Each experiment was repeated at least three times.

### 4.7. Western Blotting and Immunohistochemistry

For the following experiments, 1.0 × 10^5^/well cells were transfected with 10 nM of siRNA for 48 h using the methodology described in Section 4.2. Total protein lysates were prepared from cultured cells using NuPAGE LDS Sample Buffer (Invitrogen, Thermo Fischer Scientific, Inc., Waltham, MA, USA). The antibodies used for Western blotting and immunostaining were as follows: anti-CDK2 (1:1000; cat. no. 2546; Cell Signaling Technology, Danvers, MA, USA), anti-poly (ADP-ribose) polymerase (PARP)/cleaved PARP (1:1000; cat. no. 9542; Cell Signaling Technology), and anti-Bcl-xL (1:1000; cat. no. 2764; Cell Signaling Technology). The secondary antibodies were either peroxidase-conjugated anti-rabbit IgG (1:5000; cat. no. 7074S; Cell Signaling Technology, Inc.) or anti-mouse IgG (1:5000; cat. no. 7074S; Cell Signaling Technology, Inc.). The protein levels were quantified using ImageJ software (ver. 1.52; http://rsbweb.hnih.gov/ij/index.html, accessed on 18 October 2023) as described previously [77,78].

### 4.8. Xenograft Model

A xenograft mouse model was developed using T24 cells transfected with si*BCYRN1*-1 and siRNA-NC. Female nude mice (BALB/c nu/nu, 6 weeks old) were obtained from Charles River Laboratories (Yokohama, Japan). Transfected cells were collected, mixed with Matrigel, and transplanted subcutaneously into the backs of the mice on both sides (4 × 10^6^ cells/site). Five samples for each cell line were inoculated into the mice for this study, determined based on the Guidelines for the Welfare and Use of Animals in Cancer Research [79]. Tumor diameter and mouse weight were measured twice a week. Tumor volume was measured with calipers and calculated as v = (length × width^2^)  ×  (π/6). Thirty-one days after inoculation, all the mice were euthanized with 100% CO_2_, and tumor size was assessed. The excised specimens were embedded in paraffin and used for immunohistochemistry.

### 4.9. Immunohistochemistry

Immunohistochemistry was performed using an UltraVision Detection System (Thermo Scientific, Fremont, CA, USA) according to the manufacturer’s instructions. Primary rabbit monoclonal antibodies against Ki67 (ab92742; Abcam, Cambridge, UK) were diluted at 1:500 and incubated at 4 °C overnight. The secondary antibody was Goat anti-Rabbit IgG Antibody (H+L), Biotinylated (BA-1000; Vector Laboratories, San Francisco, CA, USA) diluted to 5 µg/mL and incubated for 30 min. Positive cells were quantified by counting six random microscopic fields. These experimental procedures were described in a previous report [80].

### 4.10. Isolation of Exosomes from the Sera of Patients with BC

Serum samples were collected from patients at Kagoshima University Hospital (Kagoshima, Japan) from June 2021 to November 2022. We collected 10 mL of blood from each patient prior to surgery (TURBT, total nephroureterectomy, or total cystectomy). To compare pre- and postsurgical samples in subsequent experiments, we utilized all the samples in our stocks from patients who underwent primary and secondary TUR using the same method. At our institution, secondary TUR is typically performed 4–6 weeks after the primary TUR. The serum samples were centrifuged twice (2000× *g*, 4 °C, 10 min) to collect the sera and remove the blood cell components and impurities. The collected serum samples were then heavily centrifuged (10,000× *g*, 4 °C, 30 min) and filtered through a 0.22-μm filter to remove the finer impurities, as previously reported [33]. We isolated exosomes from the filtered samples using an exoEasy Maxi Kit (Qiagen) according to the manufacturer’s protocol. The exosome concentration was determined by measuring the protein level using a Qubit4 Fluorometer (Thermo Fisher Scientific).

### 4.11. Statistics

All the data are representative of at least three independent experiments. The relationships between the two groups were analyzed using Mann–Whitney *U* tests. The relationships among three or more groups were analyzed using the multiple comparison test with the Bonferroni–Dunn method. The data were analyzed using R programming language (version 4.3.2, R Core Team).

## Figures and Tables

**Figure 1 ijms-25-05955-f001:**
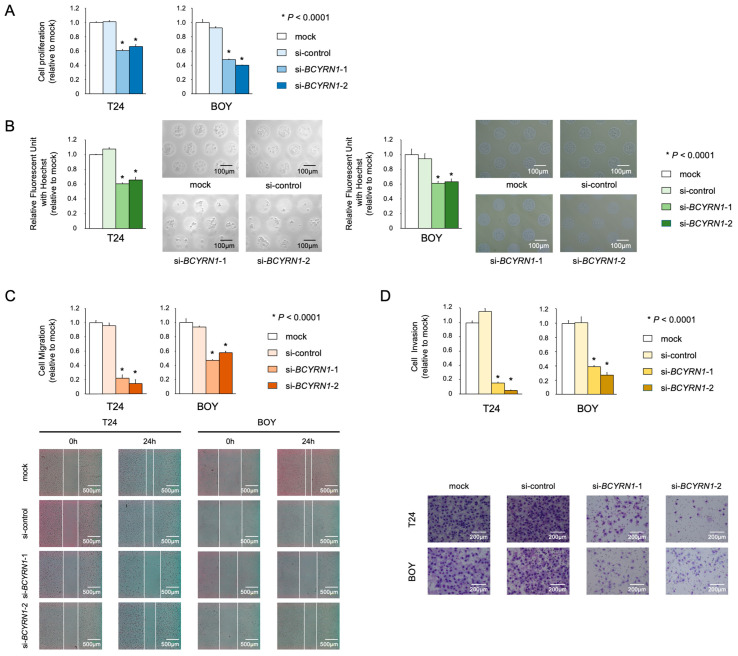
Knockdown of *BCYRN1* decreased the viability of BC cells. (**A**) Cell proliferation in two-dimensional monolayer culture was evaluated using XTT assays in BC cell lines. (**B**) Cell proliferation in 3D cell culture was evaluated using 3D spheroid assays in BC cell lines. The proliferation of BC cells was significantly reduced by the knockdown of *BCYRN1*. (**C**) Cell migration ability was measured using wound healing assays. Migration ability was significantly reduced by the knockdown of *BCYRN1*. (**D**) Cell invasion ability was measured using Matrigel invasion assays. The counted infiltrated cells were decreased in *BCYRN1*-knockdown cells compared with those in mock-transfected cells. Each experiment was repeated at least three times. The error bars indicate standard errors of the means. * *p* < 0.0001. The relationships among the four groups were analyzed using multiple comparison tests with the Bonferroni–Dunn method.

**Figure 2 ijms-25-05955-f002:**
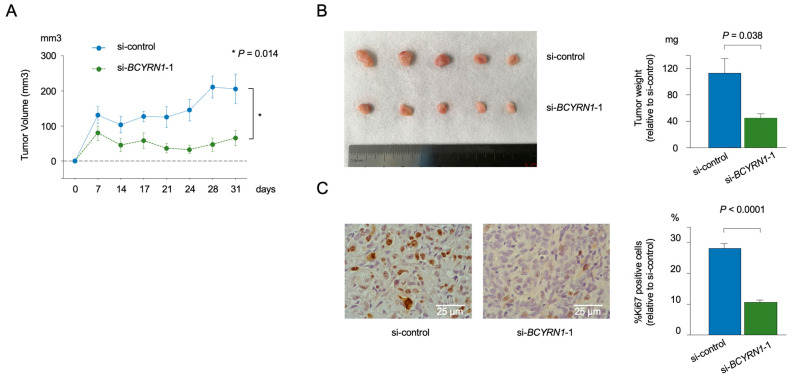
*BCYRN1* knockdown decreased BC cell viability in a xenograft mouse model. (**A**) Comparison of tumor volumes in mice subcutaneously inoculated with siNC- or si*BCYRN1*-transfected T24 cells (*n* = 5, *p* = 0.0139, Mann–Whitney *U* test). (**B**) Photograph showing excised tumor tissue from the xenograft mouse model on day 31. The graph on the right side shows significant differences between the groups (*p* = 0.038, Mann–Whitney *U* test). (**C**) Photograph showing Ki-67-positive cells in the excised tumor stained using immunohistochemistry (200×). The graph on the right side indicates the percentage of Ki-67-positive cells (*p* < 0.0001, Mann–Whitney *U* test). The error bars indicate standard errors of the means.

**Figure 3 ijms-25-05955-f003:**
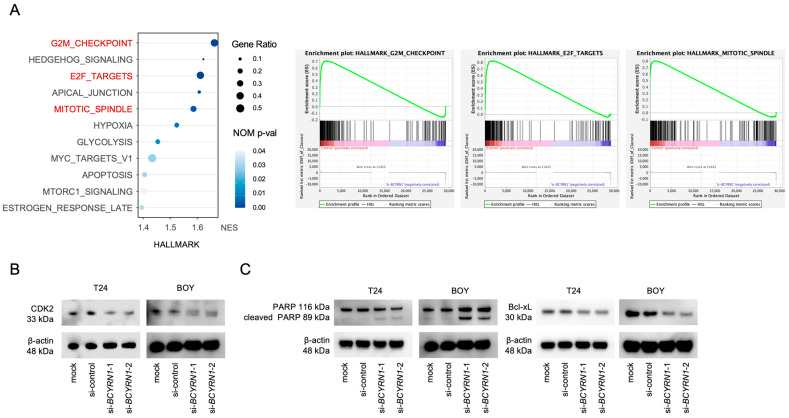
*BCYRN1* downregulation affected the cell cycle and cell apoptosis. (**A**) Hallmark gene enrichment analysis of downregulated gene expression events in T24 and BOY cells transfected with si*BCYRN1*-1. The color shows the *p*-value, the *y*-axis shows the NES, and the circle size indicates the gene ratio. The lower panel shows each of the enriched gene set events. (**B**,**C**) Western blot analysis of cleaved PARP, CDK2, and Bcl-xL expression in cells transfected with si*BCYRN1* or siNC.

**Figure 4 ijms-25-05955-f004:**
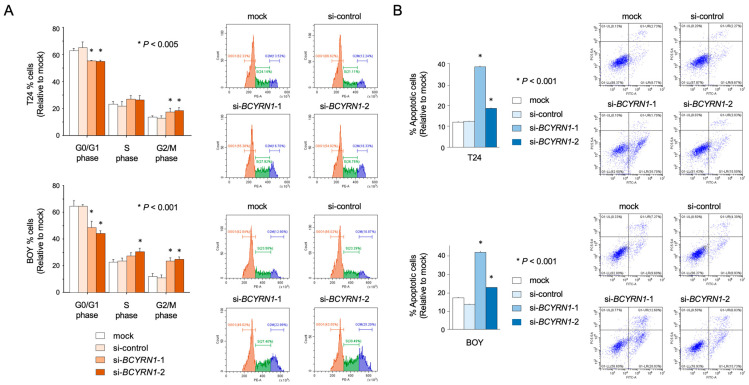
Knockdown of *BCYRN1* induced G_2_/M arrest and apoptosis in BC cells. (**A**) The percentage of cells in each cell cycle phase among groups of cells transfected with si*BCYRN1* or siNC (* *p* < 0.005). (**B**) The percentage of apoptotic cells, as determined using flow cytometry, among cells transfected with si*BCYRN1* or siNC (* *p* < 0.001). The relationships among the four groups were analyzed using multiple comparison tests with the Bonferroni–Dunn method.

**Figure 5 ijms-25-05955-f005:**
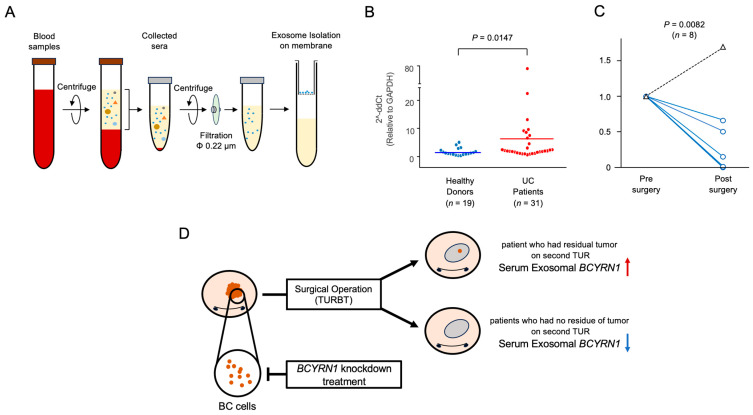
High serum exosomal *BCYRN1* levels were observed with the presence of BC tumors. (**A**) Method for the isolation of exosomes from patient serum using ultracentrifugation. (**B**) Comparison of serum exosomal *BCYRN1* expression levels between patients with BC and healthy donors (*p* = 0.0085, Mann–Whitney *U* test). The error bars indicate standard errors of the means. The horizontal bar indicates the mean of the samples. (**C**) Comparison of serum exosomal *BCYRN1* expression levels before and after TURBT in the same patient. The solid line represents patients without residual tumors, in whom expression of *BCYRN1* significantly decreased after completion, and the dashed line represents the patients with residual tumors on the second TUR. (**D**) Schema of this study.

## Data Availability

All the data generated or analyzed during this study are included in this published article.

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
