# Peer review of "LncRNA BCYRN1 as a Potential Therapeutic Target and Diagnostic Marker in Serum Exosomes in Bladder Cancer"

_ijms, 2024, doi:10.3390/ijms25115955_

Round 1
Reviewer 1 Report
Comments and Suggestions for Authors
In this manuscript titled, “The lncRNA BCYRN1 as a Potential Therapeutic Target and Diagnostic Marker in Serum Exosomes in Bladder Cancer”, the author Arima et al have investigated the role of serum exosomal BCRYN1 as a therapeutic and diagnostic marker for bladder cancer. The results suggests that knockdown of BCRYN1 result in reduction in bladder cancer cell viability and slows down tumor growth in a xenograft mouse model.
1. Introduction line 56; provide references for first sentence “ Noncoding RNAs….human genome”.
2. Line 64, “Additionally, urinary miR-96/-183 may be a diagnostic 64 marker.” provide references.
3. Fig 1B, the spheroid images for the mock and si-control group in the BOY cell line, has a different background color as compared to the siBCYRN1-1/-2 treated cells which could suggest these groups were imaged at different intensities. It will be useful to provide the spheroid images of the treatment groups imaged at the same intensity (Fig 1B, BOY cell line).
4. Caption for figure 2B: Include the scale bar and mention the magnification details in the caption.
5. Discussion: Line 88 “ BCYRN1 is known to exhibit high expression …..and glioblastoma” is repeated in the discussion section (line 205). You may consider removing this sentence from the discussion section.
6. The gene expression levels in tumors from the control and treatment group can provide useful information (Figure panel 2).
7. Materials and method section,4.6 Apoptosis and cell cycle analysis and 4.7 western blotting- the details of the number of cells used, treatment duration, siRNA dose is missing. Mention the details.
Author Response
Dear Editors and Reviewers
We appreciate your review of our manuscript and your valuable advice.
We have addressed your comments with point-by-point responses and revised the manuscript accordingly in highlight each color.
Response to the Comments by the Reviewer 1, highlighted yellow in the manuscript.
Comment 1, 2:
Introduction line 56; provide references for first sentence “Noncoding RNAs….human genome”.
Line 64, “Additionally, urinary miR-96/-183 may be a diagnostic 64 marker.” provide references.
Response 1,2:
We apologize for the lack of references.
We have added each reference to substantiate the statements in revise on Line 56-57 and 64-65.
Comment 3:
Fig 1B, the spheroid images for the mock and si-control group in the BOY cell line, has a different background color as compared to the siBCYRN1-1/-2 treated cells which could suggest these groups were imaged at different intensities. It will be useful to provide the spheroid images of the treatment groups imaged at the same intensity (Fig 1B, BOY cell line).
Response 3:
We gratefully appreciate reviewer’s comment. We have replaced to new photography data in order to improve consistency in photo intensity (Figure 1B).
Comment 4:
Caption for figure 2B: Include the scale bar and mention the magnification details in the caption.
Response 4:
We would like to thank the reviewers for their feedback.
As was pointed out, a scale bar has been added to the photo and the caption now includes a magnification factor. (Figure 2B and Line 142)
Comment 5:
Discussion: Line 88 “BCYRN1 is known to exhibit high expression …..and glioblastoma” is repeated in the discussion section (line 205). You may consider removing this sentence from the discussion section.
Response 5:
We greatly appreciate the reviewer’s comments. The text of the discussion section has been reorganized and simplified. We deleted the repeated sentence from discussion part. (Line 210-213)
Comment 6:
The gene expression levels in tumors from the control and treatment group can provide useful information (Figure panel 2).
Response 6:
We greatly appreciate the reviewer’s comments. We understand that is very useful for showing our data attracting. But regrettably, we could not perform the experiments the reviewers required, including making slides for RNA extraction from paraffinized tumor block for short period of time. Since the effect of siRNA generally lasts 5-7 days, it is expected that expression analysis using excised samples will show no difference of BCYRN1 expression between the two samples. We are very sorry that we were unable to conduct the required experiments.
Comment 7:
Materials and method section,4.6 Apoptosis and cell cycle analysis and 4.7 western blotting- the details of the number of cells used, treatment duration, siRNA dose is missing. Mention the details.
Response 7:
We greatly appreciate the reviewer’s comments. As was pointed out, we had not included the necessary information regarding the preparation of cells for transfection and FACS analysis. 1 x 105/well cells were transfected with 10 nM siRNA for 48 hours with the methodology of mentioned at section 4.2. We have incorporated the reviewer’s suggestions into the revised text. (Line 374-376, 388-389)
Reviewer 2 Report
Comments and Suggestions for Authors
In this study, authors attempted to evaluate the role of the lncRNA brain cytoplasmic RNA 1 (BCYRN1) in bladder cancer. The results showed knockdown of BCYRN1 significantly inhibited the proliferation, migration, invasion, and three-dimensional spheroid formation ability and induced apoptosis of BC cells. Higher serum exosomal BCYRN1 levels were found in patients with BC. Meanwhile, serum exosomal BCYRN1 levels were decreased in patients at complete resection of BC. This study provided the evidence that knockdown of BCYRN1 may have beneficial effects on BC therapy. However, more studies are required to confirm the study’s results. Some questions and comments were as follows:
1. si-BCYRN1-1 and si-BCYRN1-2 likely have distinct sequences, allowing them to selectively silence BCYRN1 expression. Why the authors just use si-BCYRN1-1-transfected T24 cells in xenograft mouse model, and ignore the si-BCYRN1-2 in this in vivo model? Meanwhile, throughout the discussion section, the author just discussed the role and mechanism of BCYRN1 on bladder cancer, and knockdown of BCYRN1 on potential therapeutic application. The author seems regard si-BCYRN1-1 as si-BCYRN1-2. However, the anti-cancer effect is quite different between si-BCYRN1-1 and si-BCYRN1-2, especially apoptosis inducing effect (Figure 4). If author provide more information to talk about this issue in the following discussion section, it may helpful to understand this study.
2. In Figure 5, there are just 10/31 BC patients display high level serum exosomal BCYRN1 as compared healthy donors. And one patient showed increased serum exosomal BCYRN1 at the second TUR. However, the authors mentioned that “serum exosomal BCYRN1 may be a promising diagnostic marker and therapeutic target in patients with BC”. The data in results section and description in conclusion seems not match. The authors should give illustrations.
Author Response
Dear Editors and Reviewers
We appreciate your review of our manuscript and your valuable advice.
We have addressed your comments with point-by-point responses and revised the manuscript accordingly in highlight each color.
Response to the Comments by the Reviewer 2, highlighted green in the manuscript.
Comment 1:
si-BCYRN1-1 and si-BCYRN1-2 likely have distinct sequences, allowing them to selectively silence BCYRN1 expression. Why the authors just use si-BCYRN1-1-transfected T24 cells in xenograft mouse model, and ignore the si-BCYRN1-2 in this in vivo model? Meanwhile,
throughout the discussion section, the author just discussed the role and mechanism of BCYRN1 on bladder cancer, and knockdown of BCYRN1 on potential therapeutic application. The author seems regard si- BCYRN1-1 as si-BCYRN1-2. However, the anti-cancer effect is quite different between si-BCYRN1-1 and si-BCYRN1-2, especially apoptosis inducing effect (Figure 4). If author provide more information to talk about this issue in the following discussion section, it may helpful to understand this study.
Response 1:
We greatly appreciate the reviewers for sharing valuable insight and perspective, which I find very important. We observed selective knockdown of BCYRN1 with siBCYRN1-1/-2 and higher efficacy of knockdown by transfection with siBCYRN1-1 in BC cell lines than with siBCYRN1-2 (supplementally Fig. 1). It is assumed that the high or low knockdown efficacy affects the discrepancies in the results of the loss-of-function assays in vitro. Especially, we suspect that the difference in knockdown effects between siBCYRN1-1 and siBCYRN1-2 is particularly evident in the results of the apoptosis assay. For xenograft assay in vivo, we planned xenograft experiments using only siBCYRN1-1 to reduce the number of animal casualties, which has a higher efficacy in knockdown and a stronger effect in cellular assays. We hope that the additional information provided in the text will help to explain why we used only siBCYRN1-1. (Line 125-127)
Comment 2:
In Figure 5, there are just 10/31 BC patients display high level serum exosomal BCYRN1 as compared healthy donors. And one patient showed increased serum exosomal BCYRN1 at the second TUR. However, the authors mentioned that “serum exosomal BCYRN1 may be a promising diagnostic marker and therapeutic target in patients with BC”. The data in results section and description in conclusion seems not match. The authors should give illustrations.
Response 2:
We would like to thank the reviewers for their feedback.
Please accept my apologies for the typographical error in the article. In fact, the patient whose exosomal BCYRN1 levels increased between the first and second TUR had residual tumor on the second TUR (the dashed line case in Figure 5C, n=1). We can indicate both the pathological report. The relevant text has been replaced and highlighted in green (Line 196-197, 249-251, 271-277). We added illustration to explain our results (Figure 5D).
Reviewer 3 Report
Comments and Suggestions for Authors
The authors in the present article have highlighted role of the lncRNA brain cytoplasmic RNA 1 (BCYRN1) in bladder cancer. BCYRN1 downregulation was found to inhibit the proliferation, migration, invasion, and spheroid formation ability and induced apoptosis of BC cells through G2/M arrest of the cell cycle. Clinically they showed that BCYRN1 expression in serum exosomes from BC patients was significantly higher than in healthy donors, suggesting it as a biomarker for BC.
The article is well written, but the authors must include additional experimentation to enrich the present work.
Points to be considered:
1. There are several articles that showed that downregulation of BCYRN1 leads to decreased tumorigenesis in CRC, BC, and others. To this what does this study adds upon to include any new information. Already an article was there published in 2021 showing role of exosomal BCYRN1 to drive lymphatic metastasis of bladder cancer, doi: 10.1002/ctm2.497.
2. Refer to figure 2 how stable is the siRNA mediated transient downregulation to show effect in in vivo mice model? Mostly stable knockdown generation using either shRNA or others are used for in vivo study models. Any transient downregulation does not be considered for the animal study for long times.
3. Also, the authors must show the expression levels of BCYRN1 from the control and knockdown mice set of tumor tissues at least through gene expression analysis to confirm their observation instead of looking for only Ki67.
4. The number of patients sample included in the study is not sufficient to conclude BCYRN1 expression levels to be a biomarker in BC. The authors should include more patient samples to claim this part.
5. Do the authors correlate the expression of any other known exosomal markers for BC progression in their study? There is lack of positive and negative control inclusion in this part of the study.
6. To confirm the hypothesis the authors should include a co-culture assay with the BCYRN1 expressing exosomes from patients can regulate any non-transformed cells to cancerous type. This will validate their observation more with molecular markers assay by western or phenotypic changes.
7. What will be the effect in cell cycle analysis and cell apoptosis if the BCYRN1 expressing exosomes from patients incubated with transformed and non-transformed cells?
8. The study lacks mechanistic insight. What are the other pathways that gets significantly up or down regulated upon siBCYRN1 as analyzed by the RNA seq studies. Why did the authors only focus on the cell cycle part.
9. Please discuss on the clinical importance of the study. Please focus on what makes this study more comprehensive and novel than that already reported works.
Comments on the Quality of English LanguageModerate edition
Author Response
Dear Editors and Reviewers
We appreciate your review of our manuscript and your valuable advice.
We have addressed your comments with point-by-point responses and revised the manuscript accordingly in highlight each color.
Response to the Comments by the Reviewer 3, highlighted blue in the manuscript.
The authors in the present article have highlighted role of the lncRNA brain cytoplasmic RNA 1 (BCYRN1) in bladder cancer. BCYRN1 downregulation was found to inhibit the proliferation, migration, invasion, and spheroid formation ability and induced apoptosis of BC cells through G2/M arrest of the cell cycle. Clinically they showed that BCYRN1 expression in serum exosomes from BC patients was significantly higher than in healthy donors, suggesting it as a biomarker for BC. The article is well written, but the authors must include additional experimentation to enrich the present work.
Comment 1:
There are several articles that showed that downregulation of BCYRN1 leads to decreased tumorigenesis in CRC, BC, and others. To this what does this study adds upon to include any
new information. Already an article was there published in 2021 showing role of exosomal BCYRN1 to drive lymphatic metastasis of bladder cancer, doi: 10.1002/ctm2.497.
Response 1:
We greatly appreciate the reviewer’s comments for important perspectives.
As the reviewer pointed out, the publication of exosomal BCYRN1 with BC is already exist.
Our report differs in perspective on the following points.
The previous report showed urine exosomal BCYRN1 drives lymphatic metastasis of Bladder Cancer. On the other hand, urine exosomes strongly reflect the status of neoplastic lesions in the urinary tract, urinal contamination with hematuria, pyuria, and bacteriuria causing inflammation should be ruled out. Most bladder tumors are accompanied by inflammation in the surrounding area. Whereas serum exosomal samples serum exosomal samples do not need to take this into account. Furthermore, the evaluation of its expression over time in the same patient has never been reported before. We reported decrease of cell viability with downregulation of BCYRN1 in bladder cancer cells, which might lead to therapeutic possibilities. We added the statement to discussion part, highlighted green. (Line 277-286)
Comment 2:
Refer to figure 2 how stable is the siRNA mediated transient downregulation to show effect in in vivo mice model? Mostly stable knockdown generation using either shRNA or others are used for in vivo study models. Any transient downregulation does not be considered for the animal study for long times.
Response 2:
We greatly appreciate the reviewer’s comments.
As the reviewer pointed out, the effect of si-RNA prolongs for 5-7 days. Therefore, it is plausible that long-term suppression of expression is desirable. However, we considered that xenograft assay using siRNA is also acceptable especially considering therapeutic applications using drug delivery. Because we can observe the effects, including decrease of cell proliferation in murine xenograft assay with down-regulation of the gene, even short-term suppression of gene expression. Moreover, murine xenograft assay with siRNA can be used to evaluate the effect of down-regulation in the living body. We previously have reported a Xenograft experiment in vivo using siRNA (Tamai et al. Mol Oncol 2022, 16, 1329–1346, doi:10.1002/1878-0261.13192.). Thus, we performed murine xenograft assay with siRNA in this study.
Comment 3:
Also, the authors must show the expression levels of BCYRN1 from the control and knockdown mice set of tumor tissues at least through gene expression analysis to confirm their observation instead of looking for only Ki67.
Response 3:
We greatly appreciate the reviewer’s comments. We understand that is very useful for showing our data certain. But regrettably, we could not perform the experiments the reviewers required, including making slides for RNA extraction from paraffinized tumor block for short period of time. We are very sorry that we were unable to conduct the required experiments. Since the effect of siRNA generally lasts 5-7 days, it is expected that expression analysis using excised samples will show no difference of BCYRN1 expression between the two samples. We are very sorry that we were unable to conduct the required experiments.
Comment 4:
The number of patients sample included in the study is not sufficient to conclude BCYRN1 expression levels to be a biomarker in BC. The authors should include more patient samples to claim this part.
Response 4:
We greatly appreciate the reviewer’s comment. We understand the importance of the reviewer’s concern. We added more 2 data of patient’s serum exosomal BCYRN1 expression as we could collect new samples after first submission of this manuscript. Both indicated decrease of BCYRN1 expression after complete resection of bladder tumor similar to the other patients we already reported. On the other hand, the patient who showed increase of BCYRN1 expression was still one patient. We added statement and limitation into the manuscript. (Line 18-22, 206-208, 277-286)
Comment 5:
Do the authors correlate the expression of any other known exosomal markers for BC progression in their study? There is lack of positive and negative control inclusion in this part of the study.
Response 5:
We greatly appreciate the reviewer’s comments.
As was pointed out, several lncRNAs and microRNAs are expressed within the exosomes of bladder cancer patients and have been reported to be involved in its progression. However, those reports are still scarce. Therefore, we didn’t investigate the other RNA expression but BCYRN1 as positive or negative control.
Comment 6:
To confirm the hypothesis the authors should include a co-culture assay with the BCYRN1 expressing exosomes from patients can regulate any non-transformed cells to cancerous type. This will validate their observation more with molecular markers assay by western or phenotypic changes.
Comment 7:
What will be the effect in cell cycle analysis and cell apoptosis if the BCYRN1 expressing exosomes from patients incubated with transformed and non-transformed cells?
Response 6 and 7:
We greatly appreciate the reviewers for sharing valuable insight and perspective, which we understand very important. As the reviewer pointed out, our experimental results show that BCYRN1 expression is higher in patient’s serum exosome, and it is highly likely that co-culture using them will show phenotypic changes in vitro. However, the total amount of exosomes obtained from one patient's 10ml blood collection is about 100-200 µg, and it is difficult to make a large amount of exosomes from patient’s sera to add to cell culture. Thus, we were unable to proceed with the experiment with the limited time for revise. We would like to regard this as future work.
Comment 8:
The study lacks mechanistic insight. What are the other pathways that gets significantly up or down regulated upon siBCYRN1 as analyzed by the RNA seq studies. Why did the authors only focus on the cell cycle part.
Response 8:
We greatly appreciate the reviewer’s comments.
In the present experimental data, GSEA analysis by RNA-seq showed a high Gene Ratio and several genes related to cell division and cell cycle were ranked in the low P-value gene group. Other gene sets, including Hedgehog signaling, Apical Junction, MYC targets, and some, resulted lower Gene Ratio, higher p-value, or both. These data suggest that the cell cycle is most affected by the downregulation of BCYRN1, and we focused our attention on this area. We added the statement to the manuscript. (Line 164-167)
Comment 9:
Please discuss on the clinical importance of the study. Please focus on what makes this study more comprehensive and novel than that already reported works.
Response 9:
We greatly appreciate the reviewer’s comment. We understand the importance of the reviewer’s concern. In our study, we focused our analysis on serum exosome rather than urinary exosome, which has been reported previously. We also examined how downregulation of BCYRN1 as a therapeutic target affects bladder cancer cells. Although not mentioned in the manuscript, RNA sequence showed that some miRNAs were upregulated in cells treated with siBCYRN1-1, suggesting that BCYRN1 is a sponge for these miRNAs. But we could not reproduce the obvious changes in the in vitro experiments, so further analysis could not be performed in this study. This is a weakness of this study. However, this study shows the potential for application to Drug Delivery Systems using siRNA, lncRNA and exosomes. Next, we indicated the higher expression of serum exosomal BCYRN1 in the patients who have urinary bladder tumor. This study utilized serum exosomes, which may allow evaluation of systemic tumor pathology compared to urinary exosomes. While urinary exosomes can be a useful tool for cancer detection with reflection of tumor condition in urinary tract, the contamination of inflammation, hematuria, bacteriuria, and pyuria should always be ruled out. Due to limitations of ethical review, we were only able to collect serum from patients with no metastases and only bladder tumors in the present study. However, this study may serve as a stepstone for the evaluation of systemic bladder cancer disease status, including metastases.
We added discussion statement to the discussion part of the manuscript. (Line 271-277)
Round 2
Reviewer 2 Report
Comments and Suggestions for Authors
The authors have answered all the questions.
Author Response
Dear Editors and Reviewers
We would like to express our gratitude for your decision to recommend minor revisions in the International Journal of Molecular Sciences. We have addressed additional comments with point-by-point responses and revised the manuscript accordingly in highlight each color.
The authors have answered many, but not all of Reviewer 3's questions:
Authors must also address comments 3, 5, 6, and 7 in the text and provide the underlying explanations or limitations of their study before the article can be considered for publication in IJMS.
Comment 3:
Also, the authors must show the expression levels of BCYRN1 from the control and knockdown mice set of tumor tissues at least through gene expression analysis to confirm their observation instead of looking for only Ki67.
Response 3:
We understand that the underlying explanations or statement of limitations about the reviewer’s comment are very important and need to be included in the manuscript. We have added the statement in the manuscript accordingly: line 215-217, 233-238.
Comment 5:
Do the authors correlate the expression of any other known exosomal markers for BC progression in their study? There is lack of positive and negative control inclusion in this part of the study.
Response 5:
We understand that the underlying explanations or statement of limitations about the reviewer’s comment are very important and need to be included in the manuscript. We have added the statement in the manuscript accordingly: line 268-270, 278-283.
Comment 6:
To confirm the hypothesis the authors should include a co-culture assay with the BCYRN1 expressing exosomes from patients can regulate any non-transformed cells to cancerous type. This will validate their observation more with molecular markers assay by western or phenotypic changes.
Comment 7:
What will be the effect in cell cycle analysis and cell apoptosis if the BCYRN1 expressing exosomes from patients incubated with transformed and non-transformed cells?
Response 6 and 7:
We understand that the underlying explanations or statement of limitations about the reviewer’s comment are very important and need to be included in the manuscript. We have added the statement in the manuscript accordingly: line 286-290, 306-309.
